# Sex Differences in Mate Choice Preference Characteristics of *Aequidens rivulatus*

**DOI:** 10.3390/ani12091205

**Published:** 2022-05-07

**Authors:** Haixia Li, Jie Wang, Xu Zhang, Yu Hu, Qinglin Cai, Ying Liu, Zhen Ma

**Affiliations:** 1Key Laboratory of Environment Controlled Aquaculture, Ministry of Education, Dalian 116023, China; lihaixia@dlou.edu.cn (H.L.); wangjie@dlou.edu.cn (J.W.); zhangxu@dlou.edu.cn (X.Z.); huyu@dlou.edu.cn (Y.H.); zhangjia@dlou.edu.cn (Q.C.); 2College of Marine Technology and Environment, Dalian Ocean University, #52. Heishijiao Street, Shahekou District, Dalian 116023, China; 3Southern Marine Science and Engineering Guangdong Laboratory, Guangzhou 511458, China; yingliu@dlou.edu.cn; 4College of Biosystems Engineering and Food Science, Zhejiang University, 866 Yuhangtang Road, Hangzhou 310058, China

**Keywords:** mate choice, mate preferences, no-choice test, sex differences, sex selection, *Aequidens rivulatus*

## Abstract

**Simple Summary:**

Generally, animals prefer mating with partners of the opposite sex with specific features, which suggests that animals tend to choose mates with particular phenotypic traits. However, there are some differences in mate choice behavior and criteria between males and females. This study analyzed these differences between males and females in *Aequidens rivulatus* by quantifying body size, behavioral intention, and appearance. The results showed that males paid more attention to preference degree and female attractiveness, whereas females focused on ability and physical strength displays. Consequently, males who chose to mate were primarily associated with body size, behavioral intention, and appearance, whereas the preferences of females were body size, appearance, and behavioral intention. Collectively, our initial findings revealed that males and females have different criteria for mate choice, which is vital in determining successful mating and improving artificial mating.

**Abstract:**

The mating roles of males and females, to a certain extent, are dynamic and variable. Several factors influence the mate choice process. Nonetheless, the main preference features have not yet been fully understood in *Aequidens rivulatus*. In this study, because of its natural pairing characteristics, *A. rivulatus* was selected to explore the mate choice preferences of different sexes. Specifically, male and female behavioral performances were described and quantified through a “no-choice paradigm” during mate choice. A total of 12 behavioral performances were defined in male mate choice (experiment 1), whereas 14 behavioral performances were defined in female mate choice (experiment 2). According to the obtained results, unselected females did not display any proactive behaviors in experiment 1, whereas unselected males exhibited proactive behaviors in experiment 2, including quivering, nipping, tail beating, swimming up and down, and aggression. It was also found that both male and female individuals tend to express dislike rather than like. Those behaviors with higher frequencies (e.g., quivering) often mean less energy expenditure, thus easier repeatability. Moreover, principal component analysis (PCA) was employed to extract and identify mate choice preference features. Preliminary results indicated that male preferences for a mate were mainly associated with body size, behavioral intention, and appearance, whereas the intensity of female preferences was in the order of body size, appearance, and behavioral intention. In addition, sex hormone levels were associated with mate choices.

## 1. Introduction

Sexual selection manifests in two forms: intersexual selection and intrasexual mating competition [1,2]. Until 40 years ago, studies on sexual selection only focused on intrasexual mating competition. Nevertheless, the intersexual selection of a mate is an important form of sexual selection and the evolutionary process [3]. Since Darwin’s time, females have been considered to be more dominant and active in mate choice than males, as they invested more in offspring nurturing [1,4,5,6,7]. However, emerging evidence suggests that males may be more dominant and active, even in species with little investment in offspring tending [8]. Therefore, the mating roles of males and females are dynamic and variable to a certain extent [9,10,11].

Individuals prefer mating with individuals of the opposite sex with one or several features [12,13,14,15]. This suggests that animals tend to choose mates with dominant traits that are ultimately linked to the expression of a specific gene pattern. Therefore, mate preference is an essential driver of environmental adaptation and population evolution [16,17]. Studies have also revealed that animals judge their mates based on external features [18,19]. Representative characteristics of fish mate choice may be associated with appearance and behavioral performance. Mate choice is a complex process related to communication between potential partners in many ways. That is, mate choice is a decision made after comprehensive consideration. Several studies have already demonstrated that body size, body color, and ornament may indicate the capacity of reproduction and fitness of fish [9,20,21,22]. There are also several studies showing that fish tend to choose mates that exhibit intensive courtship behaviors [23,24]. Nevertheless, the external dominant features in fish mate choice processes have not been established, which suggests that the preference features of the mate choice process are not yet clear in *Aequidens rivulatus*.

Tests for quantifying mate preferences, which can be divided into choice tests and no-choice tests, are important for evaluating sexual selection [25]. Experimental designs directly determine mating preference results. Most current research on mating preferences adopts the “selection paradigm”, specifically the dichotomous choice method. Nonetheless, given that the dichotomy focuses on selecting relative preferences for one or two given characteristics, the observed strength of preferences may be amplified [26,27]. It is worth noting that the no-choice test better explains the process of natural selection, and provides a more comprehensive and authentic understanding of differences in the intensity of sexual response preferences [28]. First, observational field results revealed that fish might not encounter multiple opposites while choosing a mate, hence preferring sequential selection [29]. Second, the no-choice test focuses on absolute preferences in mate choice. It allows for a comprehensive assessment of potential mates in the test, preventing the amplification of preferences and exhibiting their true preferences [30,31]. In fact, although the no-choice test showed much about authentic mate preferences, more no-choice tests need to be further conducted to investigate the critical points.

*A. rivulatus* (Günther, 1960) is a medium-sized fish species originating from Ecuador and Peru [32,33], feeding on worms, crustaceans, and insects [34]. It is also an important ornamental cichlid in China, with evident differences in phenotype between male and female individuals after sexual maturity. *A. rivulatu**s* exhibits natural pairing characteristics, with an evident preference for choosing a mate. Herein, we used the no-choice test to investigate differences in the behavioral performances of different sexes of *A. rivulatus* during mate choice. Next, principal component analysis (PCA) was utilized to extract and identify the primary preference features. Our findings will provide valuable insights into the behavioral performance features and gender-based differences in the mate choice process of *A. rivulatus*. In addition, it is expected that this study will unravel the evolutionary behavioral mechanisms of fish sexual choice.

## 2. Materials and Methods

This study complied with the policies relating to animal experiments (the ARRIVE and PREPARE guidelines). All procedures performed in this study were reviewed and approved by the Dalian Ocean University ethics committee (GBT 35892-2018).

### 2.1. Origin and Maintenance of Experimental Fish

Experiments were performed at the Key Laboratory of Environment Controlled Aquaculture (AET), Dalian city, Liaoning Province, China, 2020. Fish of the same genetic background were generated and raised under identical conditions. A total of 134 sexually mature, non-reproducing female and male fish (n_♀_ = 63, n_♂_ = 71) were temporarily raised in two glass tanks of similar size (120 × 55 × 47 cm; water volume = 260 L). The criteria of sexual maturity could be judged by the behavior of the fish starting to dig out the substrate. Notably, the two glass tanks were designed as recirculation systems, with a crystal sand bottom (particle size 3–4 mm), and were continuously supplied with aerated tap water. Fish were fed twice daily on commercial feed (crude protein ≥ 44% and crude fat ≥ 5%).

### 2.2. Experimental Design

The experiments commenced on 18th December 2019 and ended on 21st January 2020. Two glass tanks (55 × 55 × 47 cm) were used in the experiment, whose side and back walls were sealed with plastic sheeting to minimize external interference. We designed a casing (a drilled tube with a large tube with a hole) for aeration to prevent oxygenator bubbles from affecting image processing. Cameras were mounted directly above and in front of the tank, and videos were recorded between 19:00 and 21:00. According to our observations and literature findings, the mate choice behavior of *A. rivulatus* is more active at night than during the day, and it usually spawns at night or early morning (https://www.seriouslyfish.com/species/andinoacara-rivulatus/ accessed on 10 March 2022). Two no-choice test experiments were performed to evaluate the absolute preferences of males and females during the mate choice process. Notably, the water was replaced after every trial to prevent pheromone accumulation in the tanks. In order to facilitate statistics and improve the readability of the manuscripts, the behaviors of *A. rivulatus* were described based on the relevant references [35,36] and defined as “behavioral performance” (see Table 1). The selected individuals were used only once in all experiments.

#### 2.2.1. Experiment 1: Male Mate Choice

Fish were randomly selected and placed into a glass cylinder at 19:00 the day before the experiment. Only sexually mature males with normal behavior and nonwounded skin were used for the experiment. The cost of nesting is related to individual’s capacity to maintain and defend the nest, reflecting the competitive capacity and energetic status of the males [37,38,39]. After 24 h of male acclimatization, a randomly selected female was placed in the experimental glass tank.

Videos were recorded at first interaction between the two fish, with each video lasting 20 min. The females that made males display intense courtship behavior (including quivering, fin flickering, nipping, chafing, and following) were considered preferred individuals, even if the females were not interested in mating or rejected the approaching males, which sometimes even led to aggressive behavior of the male. If the first behavior that a male displays on a female was aggressive behavior, then the female was considered a non-preferred individual [35,36]. Females who could not be considered as preferred or non-preferred individuals within 30 min were abandoned. In experiment 1, each male was repeatedly matched with nine females; that is, a total of six males and 57 females were used (three females were abandoned). The videos from two preferred females and two non-preferred females were randomly selected for behavior analysis in each repeat test; that is, a total of 24 videos (12 videos from preferred females and 12 videos from non-preferred females) were analyzed in experiment 1.

#### 2.2.2. Experiment 2: Female Mate Choice

The female mating experiment was performed after the male mating experiment. Notably, the procedures used in this experiment were similar to those of the male mate choice experiment. Only sexually mature females with normal behavior and nonwounded skin were used for the experiment. The cost of nesting is related to individual’s capacity to maintain and defend the nest, reflecting the competitive capacity and energetic status of the females [37,38,39]. After 24 h, a randomly selected male was placed in the experimental tank. The discriminant standard was similar to that applied in Section 2.2.1. In experiment 2, each female was repeatedly matched with ten males; that is, a total of six females and 60 males were used. The videos from two preferred males and two non-preferred males were randomly selected for behavior analysis in each repeat test; that is, a total of 24 videos (12 videos from preferred males and 12 videos from non-preferred males) were analyzed in experiment 2.

### 2.3. Behavioral Observation

At the end of the experiment, the behaviors of females and males during mate choice were analyzed using Noldus EthoVision XT (version 12.0; Noldus Information Technology, Wageningen, The Netherlands). Specifically, the dynamic subtraction method was used, and the sampling rate was set at 25 frames per second. A smoothing (lowness) method eliminated small movements, including system background noise. In addition, data were checked by an all-occurrence recording method to ensure accuracy. The behaviors of both female and male fish during mate choice, including behavioral patterns and intention, were video recorded. Finally, behavioral intention was expressed as the frequency and percentage duration of a particular behavior.

### 2.4. Sampling and Analytical Methods

At the end of observation, all preferred and non-preferred fish in experiments 1 and 2 were anesthetized, followed by measuring body size, ornament, and sex hormone levels.

#### 2.4.1. Body Size Measurement

The body length, height, and circumference of the fish were measured using a tape measure at an accuracy of 0.1 cm, whereas body weights were determined using a scale of 0.01 g.

#### 2.4.2. Appearance Observation and Ornament Quantification

After the observation experiment, all preferred and non-preferred fish were captured using a professional camera, and the obtained images were then used to quantify the proportions of tail decoration area, facial area, body area, tail area, and aspect ratio.

#### 2.4.3. Determination of Sex Hormone Levels

Blood samples from preferred males (*n* = 5), preferred females (*n* = 5), non-preferred males (*n* = 5), and non-preferred females (*n* = 5) were analyzed. All samples were stored at room temperature for 20 min, and centrifuged at 3000 rpm for 20 min. Supernatants were then obtained and stored at −80 °C until further analysis. Levels of estradiol (E2) and testosterone (T) were determined using a commercial kit (Nanjing Jiancheng Institute of Biological Engineering, Nanjing, China) in accordance with the manufacturer’s protocol.

#### 2.4.4. Statistical Analyses

PCA was employed to extract and identify mate choice preferences. Kaiser–Meyer–Olkin (KMO) test statistics (>0.5) and Bartlett sphericity test (<0.01) were used as evaluation indexes to determine whether PCA could be carried out. All statistical analyses were performed using Microsoft Office Excel 2017 and SPSS (version 21.0). Student’s *t*-test analysis was used to compare the differences between the two groups. Non-parametric tests were used when the data were not normally distributed. *p* ≤ 0.05 was considered statistically significant.

## 3. Results

### 3.1. Male Mate Choice

#### 3.1.1. Behavioral Patterns of Mating

A total of 12 behaviors were observed in experiment 1 (Table 1). Behaviors shared by both males and females included quivering, fin flickering, nipping, mate chafing, and aggression. Behaviors specific to males included lateral display, following, tail beating, and chasing. In contrast, behaviors specific to females included threatening, freezing, and swimming up and down (Table 1). The behaviors of preferred females included nipping, quivering, fin flickering, mate chafing, and swimming up and down. Additionally, preferred females exhibited aggressive and threatening behaviors, indicating dominance. Although not all behaviors were observed in every group, preferred females displayed a high frequency of courtship behavior, whereas non-preferred females did not show any proactive behaviors and only exhibited freezing behaviors. Most males were active when presented with a preferred female, and nine behaviors, including quivering and fin flickering, were observed (Table 1). In contrast, when confronted with unattractive females, only six behaviors were observed, and they occurred less frequently.

#### 3.1.2. Behavioral Intention

In experiment 1, males demonstrated a significant difference in the intensity of their responses to preferred and non-preferred females. The results showed that the intensity of male behaviors toward their preferred females was significantly higher than that toward non-preferred females with regard to frequency and duration of quivering, lateral display, fin flickering, and following (Frequency: quivering *t*_11.01_ = 3.75, lateral display *t*_11.96_ = 4.70, fin flickering *t*_11_ = 3.129, following *t*_11.12_ = 4.20, *p* < 0.01, Figure 1a; Duration: quivering *t*_11.00_ = 3.87, lateral display *t*_13.12_ = 4.56, fin flickering *t*_11.00_ = 2.76, following *t*_11.09_ = 3.72, *p* < 0.01, Figure 1b). Regarding the frequency and duration of behaviors, including chasing and aggression, the intention of male behaviors toward their preferred females was significantly lower than that toward their non-preferred females (Frequency: chasing *t*_14.10_ = −7.90, aggression *t*_11.86_ = −6.21, *p* < 0.01, Figure 1a; Duration: chasing *t* _11.50_ = −4.79, aggression *t*_11.98_ = −4.86, *p* < 0.01, Figure 1b). Nevertheless, no significant differences were noted in nipping, mate chafing, and tail beating behaviors (*p* > 0.05; Figure 1a,b). 

Experiment 1 showed that the behavioral intensities differed between preferred and non-preferred females. The frequency of quivering was significantly higher among preferred females than that of non-preferred females (*t*_11_ = 2.43, *p* < 0.05, Figure 1c). In contrast, the frequency of freezing among preferred females was significantly lower than for non-preferred females (*t*_11_ = −4.43, *p* < 0.01, Figure 1c). Notably, we found no significant differences in other behavioral indices (*p* > 0.05, Figure 1c). With regard to behavioral durations, freezing in non-preferred females was significantly longer than for preferred females (*t*_112_ = −5.05, *p* < 0.05, Figure 1d), but there were no significant differences in other behavioral indices (*p* > 0.05, Figure 1d). 

#### 3.1.3. Behavior Performance PCA

The KMO value was 0.62 (Appendix A), indicating satisfactory suitability for PCA analysis. We retained factors with eigenvalues greater than 1 (Appendix A) and applied the varimax rotation to facilitate the interpretation of the loadings. Table 2 shows the behavioral performance of preferred female fish in experiment 1. In both steps, the first variable factor (VF1) accounted for 59.32% of the total variation, and quivering, nipping, and swimming up and down were positively loaded on the first component (PC1) (>0.90), indicating that VF1 represented self-display of preference. It should be noted that a high PC1 score reflected a high female propensity to attract males. The second variable factor (VF2) accounted for 32.01% of the total variation, and mate chafing was positively loaded on the second component (PC2) (>0.90). Thus, VF2 indicated the courtship action of female fish to male fish when they meet, reflecting the propensity of female fish to express a preference for male fish. Notably, the behaviors of non-preferred females were not subjected to PCA because non-preferred females only had freezing behavior.

#### 3.1.4. Body Size

In experiment 1, the body length, height, and circumference of preferred females were significantly smaller than those of non-preferred females (body length: Z Test, *W*_29.94_ = 60.50, *p* < 0.05; height: Z Test, *W*_29.40_ = 47.50, *p* < 0.05; circumference: Student’s t test, *t*_30.62_ = −2.66, *p* < 0.05; Figure 2a).

#### 3.1.5. Appearance and Ornamental Conditions

In experiment 1, no significant differences were found between preferred females and non-preferred females with regard to the proportion of yellow tail ornament area and aspect ratio (the proportion of yellow tail ornament area: Z Test, *W*_30_ = 127.00, *p* > 0.05; aspect ratio: Z Test, *W*_30_ = 151.00, *p* > 0.05, Figure 3a). The percentages of face and tail areas were significantly smaller in preferred females than in non-preferred females (facial area: Student’s t test, *t*_14_ = −4.56, *p* < 0.01; tail area: Student’s t test, *t*_14_ = −2.98, *p* < 0.01, Figure 3a). Moreover, preferred females exhibited significantly higher percentages of body areas than non-preferred females (body areas: Z Test, *W*_30_ = 228.00, *p* < 0.01, Figure 3a).

#### 3.1.6. Sex Hormone Levels 

In experiment 1, estradiol levels were significantly inhibited in preferred females than in non-preferred females (Student’s t test, *t*_8_ = −2.53, *p* < 0.01, Figure 4a).

#### 3.1.7. PCA Analysis of Main Preference Features

Based on the above findings, PCA was performed on all preferences of male fish in mate choice (Table 3). The results showed that VF1 accounted for 28.97% of the total variation, and body height and circumference were positively loaded on the first component (PC1) (>0.90), suggesting that VF1 mostly represented the body size of the female. VF2 accounted for 28.32% of the total variation, and quivering, nipping, and swimming up and down were positively loaded on the second component (PC2) (>0.90). Thus, VF2 was interpreted as a female courtship behavior intention. In addition, the third variable factor (VF3) accounted for 24.17% of the total variation, and the percentage of body area was negatively loaded on the third component (PC3) (<−0.90), representing female appearance conditions.

### 3.2. Female Mate Choice

#### 3.2.1. Behavioral Patterns of Mating

A total of 14 behaviors were observed in experiment 2 (Table 4). Unlike in experiment 1, the two most common behaviors in experiment 2 increased frontal biting and swimming. Behaviors common between males and females included quivering, fin flickering, nipping, following, mate chafing, tail beating, accompanying swimming, chasing, aggression, frontal biting, and swimming up and down. Notably, threatening behavior was specific to females, whereas lateral display and freezing were specific to males (Table 4). 

Only six behaviors were observed when presented with unattractive mates, and occurred less frequently. On the other hand, females displayed nine behaviors when integrated with preferred males, including quivering, fin flickering, nipping, mate chafing, aggressive, and threatening behaviors. When presented with non-preferred males, females showed eight behaviors, with all groups showing aggressive and threatening behaviors (Table 4).

#### 3.2.2. Behavioral Intention

In experiment 2, females showed a significant difference in behavioral intention between preferred and non-preferred males. Results showed that the intention of female behaviors toward their preferred males was significantly greater than that toward their non-preferred males with regard to the frequency of quivering and nipping (quivering: *t*_11.41_ = 2.72, nipping: *t*_11.06_ = 2.11, *p* < 0.05, Figure 5a). Regarding the frequency of aggression, threatening, chasing, and frontal biting, the intention of female behaviors toward their preferred males were significantly lower than that toward their non-preferred males (aggression: *t*_11.20_ = −4.12, threatening: *t*_12.33_ = −4.87, chasing: *t*_11_ = −2.30, frontal biting: *t*_11_ = −2.22, *p* < 0.05, Figure 5a). Differences in all other behavioral indices were insignificant (*p* > 0.05, Figure 5a). The period of female behaviors toward their preferred males with regard to quivering was longer than that toward their non-preferred males (*t*_11.11_ = 2.62, *p* < 0.05, Figure 5b). However, female behaviors toward preferred males in terms of aggression, threatening, and chasing were significantly shorter than that toward their non-preferred males (aggression: *t*_11.12_ = −4.10, chasing: *t*_11_ = −2.12, threatening: *t*_12.13_ = −5.07, *p* < 0.05, Figure 5b). Notably, no significant differences were found in all other behavioral indices (*p* > 0.05, Figure 5b). 

Overall, the intensities of female behaviors in female mate choice were lower than those in male mate choice. In experiment 2, frequencies of quivering, following, and lateral display behaviors were significantly higher in preferred males than non-preferred males (quivering: *t*_11.15_ = 2.20, following: *t*_11_ = 2.93, lateral display: *t*_11.00_ = 3.45, *p* < 0.05, Figure 5c). The frequency of freezing was significantly lower than that of non-preferred males (*t*_11_ = −2.43, *p* < 0.05, Figure 5c). Differences in other behavioral indices were insignificant (*p* > 0.05; Figure 5c). Regarding their durations, lateral display behaviors of preferred males were significantly longer than those of non-preferred males (*t*_11_ = 3.03, *p* < 0.05, Figure 5d). Differences in all other behavioral indices were insignificant (*p* > 0.05, Figure 5d).

#### 3.2.3. PCA for Behavioral Performance

Table 5 shows the behavioral performance features of the preferred male fish. Results showed that VF1 accounted for 26.04% of the total variation, and nipping was positively loaded on the first component (PC1) (>0.90), indicating that VF1 mostly reflected male capacity to modify the nest. VF2 accounted for 24.87% of the total variation, and mate chafing and aggression were positively loaded on the second component (PC2) (>0.90). Thus, VF2 was interpreted as a male proactive response to female encounters. In addition, VF3 accounted for 21.06% of the total variation, and quivering was positively loaded on the third component (PC3) (>0.90). Therefore, VF3 was interpreted as a male self-display. For non-preferred males (Table 6), VF1 accounted for 24.38% of the total variation, and chasing and swimming up and down were positively loaded on the first component (PC1) (>0.90), indicating that VF1 mostly represented male swimming behaviors. Notably, a high PC1 score reflected the ability of males to engage in physical activity. VF2 accounted for 24.21% of the total variation, and nipping and aggression were positively loaded on the second component (PC2) (>0.90). Thus, VF2 was interpreted as an activity of the male mouth. Moreover, VF3 accounted for 22.47% of the total variation, and tail beating was positively loaded on the third component (PC3) (>0.90), which suggests that VF3 reflected the propensity of males to hit each other.

#### 3.2.4. Body Size

Results revealed that the body length, height, and circumference of preferred males were not significantly different from those of non-preferred males (body length: Student’s t test, *t*_54_ = 1.03, *p* > 0.05; height: Z Test, *W*_54_ = 454.50, *p* > 0.05; circumference: Student’s t test, *t*_54_ = 1.03, *p* > 0.05; Figure 2b).

#### 3.2.5. Appearance and Ornamental Conditions

In experiment 2, the proportion of yellow tail ornament area in preferred males was significantly higher than for non-preferred males (Z Test, *W*_14_ = 52.00, *p* < 0.05, Figure 3b). No significant differences were observed between preferred males and non-preferred males with regard to proportions of facial area, body area, tail area, and aspect ratio (facial area: Z Test, *W*_14_ = 41.00, *p* > 0.05; body area: Student’s *t* test, *t*_14_ = 0.01, *p* > 0.05; tail area: Student’s t test, *t*_14_ = −2.10, *p* > 0.05; aspect ratio: Student’s t test, *t*_14_ = 1.59, *p* > 0.05; Figure 3b).

#### 3.2.6. Sex Hormone Levels

In experiment 2, no significant differences in testosterone levels were observed between the preferred males and the non-preferred males (Student’s t test, *t*_11_ = 1.15, *p* > 0.05, Figure 4b).

#### 3.2.7. PCA Analysis of Main Preference

Similarly, PCA was performed on all preference features of females in mate choice (Table 7). According to the obtained results, VF1 accounted for 29.42% of the total variation, and body circumference was positively loaded on the first component (PC1) (>0.90). Thus, VF1 was interpreted as a male body size. VF2 accounted for 27.96% of the total variation, and body area percentage was positively loaded on the second component (PC2) (>0.90), indicating that VF2 was interpreted as female appearance conditions. Besides, VF3 accounted for 20.49% of the total variation, and nipping behavior was negatively loaded on the third component (PC3) (>0.90), representing the intensity of female courtship behaviors.

## 4. Discussion

### 4.1. Differences in Behavioral Performance

In this work, male mate choice preferences were expressed through behaviors that included rejecting or accepting courtship by females, choosing to pursue certain females, and increasing courtship intensity. Regarding mate choice, significant individual differences were noted between males and females, which is consistent with findings of previous studies that males and females can adjust their courtship performance based on each other’s reactions [40,41]. In addition, specific courtship behaviors can be applied to specific potential mates, suggesting that courtship and mate choice are involved in mate assessment. We found that non-preferred females did not show any proactive behaviors during the male mate choice process, but only exhibited reactive freezing behaviors, which occurred in all groups. In contrast, when females were choosing their mates, non-preferred males displayed reactive freezing behaviors, as well as proactive behaviors, including quivering, nipping, tail beating, swimming up and down, and aggression. Therefore, the behaviors of non-preferred females and males during the mate choice process were inconsistent, which could be attributed to the fact that males have a weaker preference for non-preferred females and are unwilling to pursue females at a high cost [42]. This suggests that when males show a degree of preference for females, and females do not reciprocate, the energy and time of males limits their mating opportunities and brings about changes in the male mating intentions, resulting in mate choice failure and other factors repulsive behaviors such as aggression [43].

Since Darwin’s time (1859), certain sex traits have evolved through mate choice. These traits could be an expression of a male body and favorable genetic qualities, including ornament or self-display [44], and they are performed through various behaviors. Given that this presentation requires enormous energy and repetitions, quick and skillful movements will reflect the physical qualities of a partner [45]. This explains why the most intense behaviors are those with high repeatability, including dithering. In contrast, less intense behaviors are characterized by high energy consumption and low repeatability, including body friction and the chasing of preferred objects. It is worth noting that freezing was the strongest behavior of non-preferred individuals for both male and female mate choices. Interestingly, both male and female individuals tend to express dislike rather than like. In the dichotomous-choice test, attention is paid to the expression of like and is quantified by the time of preference zones, excluding the behavior of expressing dislike in mate choice [46]. Herein, the degree of preference and non-preference in mate choice can be simultaneously analyzed. Moreover, *A. rivulatus* are not monogamous cichlids, and the intensities of mate preference could be different from monogamous cichlids [47,48,49].

The PCA results showed that males concentrated on female attractiveness and the degree of preference in male mate choice, whereas females focused on practical ability and physical strength in female mate choice. This could be attributed to male preferences being based on the direct benefits associated with female fertility. In contrast, female preferences are mainly based on the benefits provided by males, including protection of juvenile fish and territory, and increased survival rates of the offspring [50,51]. 

### 4.2. Sex Hormone Levels

A previous study reported a close relationship between mate choice and mating behavior in sexual selection, indicating that sex steroid hormones are important [52]. Hormonal studies suggest that sex hormones influence courtship levels in different species [52,53]. Increased sex hormone levels cause higher courtship intensities. Steroid hormones in preference may primarily play a role in mate motivation, i.e., making males or females more likely to mate with certain partners of the opposite sex [54]. Estradiol levels in females are closely associated with reproductive status. 

Nevertheless, our analysis of male mating choices performed in this study revealed that preferred females had lower estradiol levels than non-preferred females. In general, estradiol is a typical estrogenic hormone that is commonly used to characterize the reproductive capacity of female animals. However, some studies have reported a reduction in estradiol levels during the reproduction of fish [55,56]. Egg viability did not change when the female rainbow trout was exposed to estradiol, indicating that egg maturation processes were not influenced by estradiol [57]. In addition, this may be also attributed to the fact that the females with elevated estradiol levels are always more inclusive and active, and cannot stay in one place long enough for males to recognize or be familiar with [54]. On the other hand, analysis of female mate choice revealed that there was no significant difference in testosterone levels between preferred males and non-preferred males. This is possibly because the release of male testosterone varies with female reproductive status and is unrelated to male size [58]. Sex hormone levels were not included in the PCA analysis of preference characteristics. This is because sex hormone levels indicate fish reproductive status. Given that both males and females used in this study were under reproductive status, sex hormone levels were no longer analyzed as a preference factor [59]. As the main purpose of this study was to identify the mating preferences of *A. rivulatus*, we tried to achieve this without damaging the fish through observations of appearance and behavior, while sex hormones were excluded.

### 4.3. PCA Results

Studies on mate choice behaviors have mainly focused on the ornamental display of mate choice, including body size and ornaments, as well as the evolution of fish behaviors. This study found that the order of characteristics affecting male mate preferences was body size > behavioral intention > appearance, whereas the order of characteristics influencing female mate preference was body size > appearance > behavioral intention. Body size was the primary factor affecting mate choice preference regarding PCA analysis, regardless of whether it was female or male mate choice. Body size modulates mate choice and is closely associated with the reproductive capacity of fish [60]. Generally, larger individuals are preferred. In male mate choice, female size typically indicates the egg-carrying capacity, whereas in female mate choice, the size of a male is typically indicative of its ability to protect the territory, thereby improving offspring survival [61,62]. Unlike our results, most studies on mate preferences in fish have reported a preference for larger body sizes for both male and female fish in *Cichlasoma nigrofasciatum*, *Poecilia reticulata*, and *Pelvicachromis taeniatus* species [63,64,65,66]. 

Furthermore, we discovered that males had a clear preference for smaller females, whereas males clearly preferred females with larger body areas, and females had no evident preference for male body size. The possible reason is that the size of the body area of females may affect the egg-carrying capacity and fecundity [64,65]. However, the relationships between female body size and fecundity are not consistent across species [66]. The size of the female body area might be considered an indicator for male mate choice in *A. rivulatus*. Second, the reason could be that females could have lower body size requirements than males, implying that males could be picky [1]. However, females are less picky about male body size, so the small difference in male body size may not matter. There was no increased intrasexual competition or predator factors. Thus, the female preference for the capacity to protect offspring and territory from larger males was unapparent.

Mate choice involves many factors in multiple sensory modalities and is thought to modulate changes in ornamental traits in fish [3,67]. Unique ornamental traits of fish are likely to develop through a long-term mate choice process, and thus, different ornamental traits could play a role in mate choice by attracting the opposite sex [68]. We discovered that the effective index of body size might not only be the body length index [69]; however, the body proportion of fish can be a more objective and specific index [70]. Male preference for female appearance was the weakest; however, male appearance was second only to body size in female mate choice. This could be because males attach more importance to female reproductive capacity for immediate benefits [9], whereas females focus on the body proportion of males, which could confer a genetic benefit [43].

In contrast with the findings of Berglund and Rosenqvist, our results suggest that males have a low preference for ornament size of the caudal fin when choosing a mate but a higher preference for females with a smaller proportion of facial and caudal fins, as well as a more significant body proportion [71]. This may be because caudal fin ornament is expressed in both males and females, decreasing the preference for tail ornament in males [9]. Consequently, males pay more attention to female reproductive capacities. In contrast, females preferred males with larger caudal fin decoration sizes and no preference for body proportion, which was consistent with the body size index results. This is consistent with Gronell [72], indicating that female mating success could be attributed to the orange area of the male caudal fin. Further, we predicted that male caudal fin color might affect female mate choice. Sexual selection may be a driver of speciation, with females selecting males based on color, causing reproductive isolation in sympatric but with color-divergent species [73]. A previous study on *Pundamilia pundamili* and *Pundamilia nyererei* found that female preference for male marital color leads to mating classification by females, which plays a vital role in the origin and maintenance of reproductive isolation [74].

Furthermore, the PCA results showed that males and females attached different importance to behavioral intensities. Specifically, males attached secondary importance to behavioral intensities after body size, whereas females attached the least importance to behavioral intention. This could be because female behavioral intensities were less intense than males in male mate choice. Female behaviors could indicate comfort, a strategy for examining a mate, and could also induce male courtship [75]. Thus, males pay more attention to female behaviors when choosing a mate. On the other hand, in female mate choice, the intensities of male behaviors represent a form of female harassment [76]. Moreover, females block males from increasing intensities of male courtship via behaviors, resulting in females attaching less emphasis on the intention of male behaviors when choosing a mate. However, the results of our study are only a preliminary exploration. Questions such as how *A. rivulatus* adjusts its behaviors to promote successful mate choice depending on the interaction with the potential partners remain to be more decisively demonstrated. We speculate that behavioral interactions may not alter courtship results; however, they do change the patterns and intensity of courtship behavior. We will clarify this in future research.

## 5. Conclusions

In conclusion, differences in intensities and types of courtship exist between males and females, and both male and female individuals tend to express dislike rather than like. The order of influencing characteristics for male mate preferences is body size > behavioral intention > appearance, whereas the order of influencing characteristics for female mate preferences is body size > appearance > behavioral intention.

## Figures and Tables

**Figure 1 animals-12-01205-f001:**
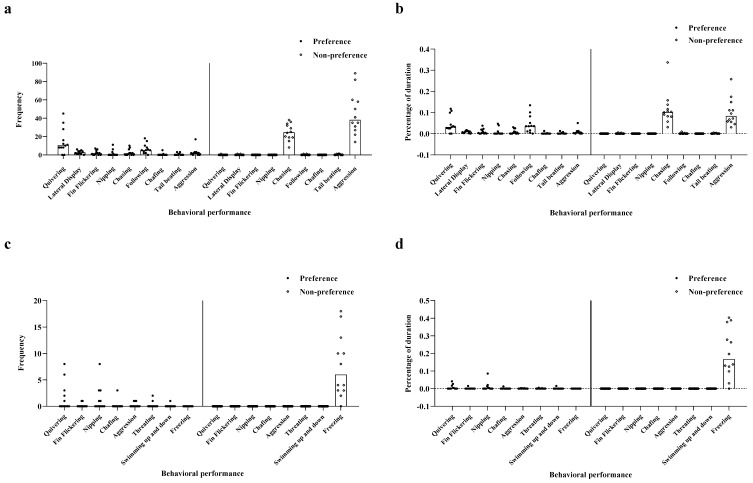
The behavioral intention of male and female fish in Experiment 1. (**a**) Frequency of male’s behavioral performance; (**b**) Proportion of duration of behavioral performance in males; (**c**) Frequency of female’s behavior performance; (**d**) Proportion of the duration of behavior performance in females.

**Figure 2 animals-12-01205-f002:**
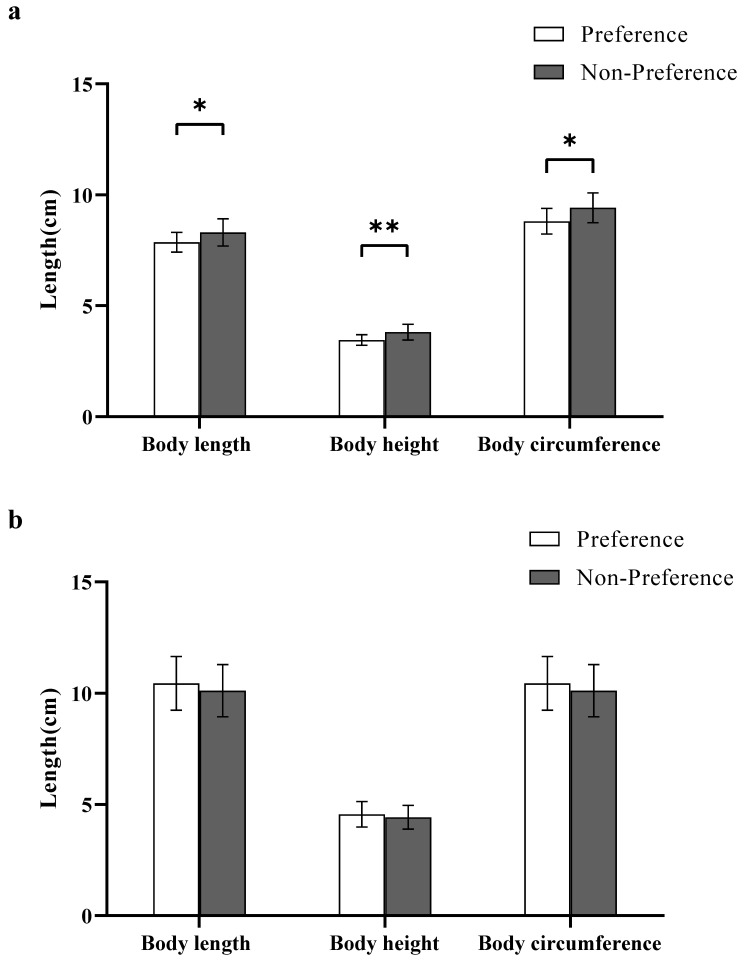
Body size preference for the different sexes. (**a**) Female body size preference in Experiment 1; (**b**) Male body size preference in Experiment 2. A significant difference is denoted by an asterisk.

**Figure 3 animals-12-01205-f003:**
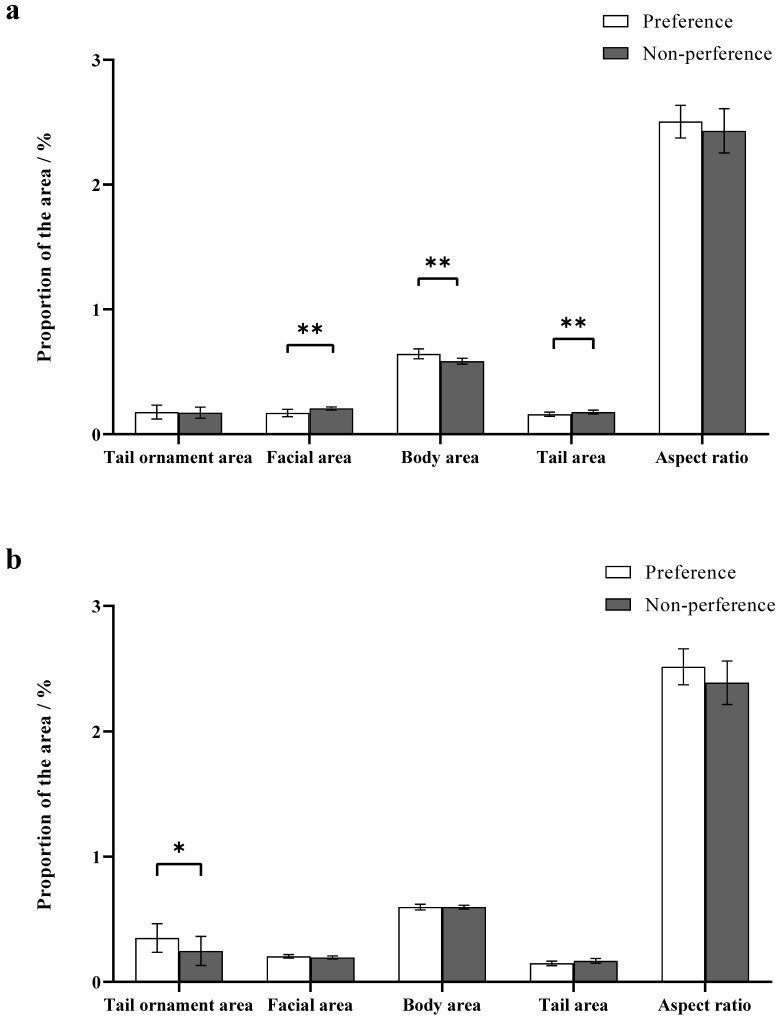
The ornamental preference for the different sexes. (**a**) Male ornamental preference in Experiment 1; (**b**) Female ornamental preference in Experiment 2. A significant difference is denoted by an asterisk.

**Figure 4 animals-12-01205-f004:**
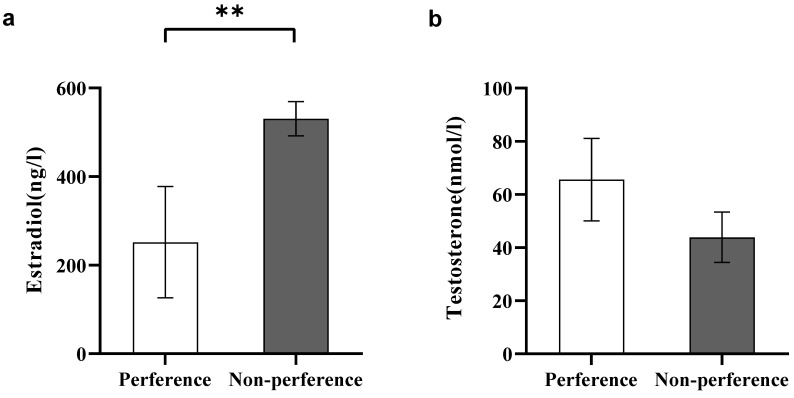
Comparison of sex hormone levels. (**a**) Estradiol levels in females in Experiment 1; (**b**) Testosterone levels in males in Experiment 2. A significant difference is denoted by an asterisk.

**Figure 5 animals-12-01205-f005:**
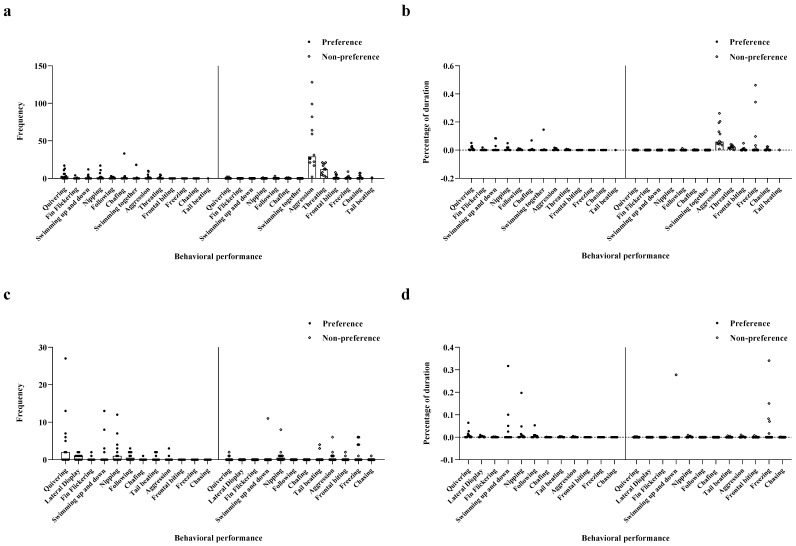
The behavioral intention of male and female fish in Experiment 2. (**a**) Frequency of female’s behavioral performance; (**b**) Proportion of the duration of behavioral performance in females; (**c**) Frequency of male’s behavior performance; (**d**) Proportion of the duration of behavior performance in males.

**Table 1 animals-12-01205-t001:** Behavioral patterns of males and their preferred and non-preferred mates.

Behavioral Patterns	Definition	Number of Groups (*n* = 6)			
		Preferred		Nonpreferred	
		Male	Female	Male	Female
Quivering	Rapid vibration of the body.	6	3	1	0
Fin Flickering	Folding and unfold the pelvic fins quickly, usually several times in a row.	5	2	0	0
Nipping	Picking up some substrate from near the nest, clean it up and spit it out.	3	4	0	0
Lateral Display	Showing one side of the body to the subject fish.	6	0	1	0
Chafing	Approaching mate caudally or medially, the two fishes contact laterally and swim together.	2	1	0	0
Following	The target fish approaches actively and swims without acceleration in the same direction with one in front and one behind.	6	0	2	0
Tail beating	Swing the tail fin towards the other fish.	4	0	3	0
Chasing	Swimming quickly to the opponent and do not return, the opponent fish speed away.	4	0	6	0
Aggression	Swimming quickly to the target fish and accompany the biting action.	6	3	6	0
Threating	Face the opponent fish with the gill cover open, the pelvic and dorsal fins spread.	0	2	0	0
Freezing	Holding the fins and body still and do not swim.	0	0	0	6
Swimming up and down	The target fish swims up and down the walls of the tank.	0	1	0	0

**Table 2 animals-12-01205-t002:** PCA outcomes of preferred females in experiment 1.

Variables	VF1	VF2
Behavioral performance of the preferred females		
Quivering	0.92	0.51
Threating	0.81	0.49
Aggression	0.37	0.86
Chafing	−0.13	0.97
Nipping	0.99	0.07
Swimming up and down	0.96	0.04
Rotation sums of squared loadings		
Total	3.56	1.92
% Total variance	59.32	32.01
Cumulative % variance	59.32	91.33

**Table 3 animals-12-01205-t003:** The main preference features of males in mate choice in *A. rivulatus*.

Variables	The Main Preference of the Males		
	VF1	VF2	VF3
Quivering	−0.05	0.93	−0.16
Nipping	−0.04	0.99	−0.11
Chafing	−0.30	−0.12	−0.23
Swimming up and down	−0.01	0.95	−0.02
Body length	0.89	−0.04	0.20
Body height	0.93	−0.13	0.24
Body circumference	0.95	−0.05	0.03
The proportion of the facial area	0.39	−0.03	0.83
The proportion of the body area	−0.29	0.10	−0.94
The proportion of the tail area	−0.03	−0.22	0.82
Rotation sums of squared loadings			
Total	2.90	2.83	2.42
% Total variance	28.97	28.32	24.17
Cumulative % variance	28.97	57.29	81.46

**Table 4 animals-12-01205-t004:** Behavioral patterns of females and their preferred and non-preferred mates.

Behavioral Patterns	Definition	Number of Groups (*n* = 6)			
		Preferred		Nonpreferred	
		Female	Male	Female	Male
Quivering	Rapid vibration of the body.	6	5	3	3
Lateral Display	Showing one side of the body to the subject fish.	0	4	0	0
Fin Flickering	Folding and unfold the pelvic fins quickly, usually several times in a row.	2	2	0	0
Nipping	Picking up some substrate from near the nest, clean it up and spit it out.	6	5	1	5
Following	The target fish approaches actively and swims without acceleration in the same direction with one in front and one behind.	5	4	1	0
Chafing	Approaching mate caudally or medially, the two fishes contact laterally and swim together.	5	1	2	0
Tail beating	Swing the tail fin towards the other fish.	0	3	1	2
Chasing	Swimming quickly to the opponent and do not return, the opponent fish speed away.	0	0	4	1
Swimming together	The relative position remains the same as the two fish swim.	2	2	0	0
Aggression	Swimming quickly to the target fish and accompany the biting action.	5	2	6	4
Frontal biting	The two fish attack mouth to mouth, biting the upper or lower lip of the opponent.	0	0	4	2
Threating	Face the opponent fish with the gill cover open, the pelvic and dorsal fins spread.	4	0	6	0
Freezing	Holding the fins and body still and do not swim.	0	0	0	5
Swimming up and down	The target fish swims up and down the walls of the tank.	3	3	0	1

**Table 5 animals-12-01205-t005:** PCA outcomes of preferred males of experiment 2.

Variables	VF1	VF2	VF3
Behavioral performance of the preferred males			
Quivering	0.09	−0.17	0.90
Lateral Display	0.89	−0.13	−0.21
Fin Flickering	0.79	0.52	−0.04
Swimming up and down	0.09	−0.37	−0.45
Nipping	0.94	−0.06	0.11
Following	−0.14	−0.08	−0.12
Chafing	0.53	0.94	−0.07
Tail beating	−0.07	−0.20	0.89
Aggression	0.05	0.92	−0.13
Rotation sums of squared loadings			
Total	2.34	2,24	1.90
% Total variance	26.04	24.87	21.06
Cumulative % variance	26.04	50.92	71.98

**Table 6 animals-12-01205-t006:** PCA outcomes of non-preferred males in experiment 2.

Variables	VF1	VF2	VF3
Behavioral performance of the non-preferred males			
Quivering	−0.18	−0.38	0.29
Swimming up and down	0.96	0.03	−0.03
Nipping	−0.05	0.91	0.15
Tail beating	−0.08	−0.05	0.95
Aggression	−0.01	0.94	0.11
Frontal biting	−0.04	0.16	0.82
Freezing	−0.20	−0.27	−0.30
Chasing	0.97	0.20	−0.06
Rotation sums of squared loadings			
Total	1.95	1.94	1.80
% Total variance	24.38	24.21	22.47
Cumulative % variance	24.38	48.59	71.06

**Table 7 animals-12-01205-t007:** The main preference features of females in mate choice in *A. rivulatus*.

Variables	The Main Preference of the Females		
	VF1	VF2	VF3
Quivering	−0.21	−0.23	0.89
Nipping	−0.07	0.04	0.93
Chafing	−0.61	−0.06	−0.14
Swimming up and down	0.35	0.56	−0.35
Body length	0.81	0.25	−0.22
Body height	0.84	0.31	−0.33
Body circumference	0.90	0.07	−0.28
The proportion of the facial area	0.40	0.84	0.02
The proportion of the body area	−0.27	−0.95	0.03
The proportion of the tail area	−0.12	0.82	−0.09
Rotation sums of squared loadings			
Total	2.94	2.80	2.05
% Total variance	29.42	27.96	20.49
Cumulative % variance	29.42	57.38	77.87

## Data Availability

The data reported within this study are available from the corresponding author upon reasonable request.

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
