# Peer review of "Sex Differences in Mate Choice Preference Characteristics of Aequidens rivulatus"

_animals, 2022, doi:10.3390/ani12091205_

Round 1

Reviewer 1 Report

The authors have presented a study in sexual selection of a fresh-water fish commonly called the Green Terror and whose natural distribution in the wild is from South America. The experiment was well thought out and implemented and the statistics and bibliography are appropriate. However, the language must be edited for flow and grammar. - makes reading the paper very difficult. Also, I think the authors should present a paragraph on the natural history and geographic distribution of the study species in the wild. Some of the figures especially 1 and 5, are of poor quality and illegible.

Author Response

The response to reviewer #1 was uploaded as separate attachment.  

Reviewer 2 Report

The paper is very poorly written, a profound English revision is needed and the experimental design presents many weaknesses and unclear parts that need adequate clarification.

Lines 13-14: Why the use of past tense? I would replace “tended” with “tend” and “were” with “are” since the concepts expressed in these sentences apply to the present.

Lines 17-18: “male preference” instead of “males that preferred”.

Lines 19-20: unclear sentence.

Line 20: What does behavioral intensity mean?

Lines 23-25: I would suggest the authors to avoid the use of “the main preference characteristics have not been established”, since the characteristics which influence mate choice preferences are likely to change among different taxa and I think it would hardly be possible to draw establish general conclusions.

Line 26: “free choice reproductive characteristics” what does it mean?

Lines 28-33: The authors must clarify this part. What is a “behavioral performance”? What are the “active sexual behaviors”? Also the meaning of “desire to express "aversion" was stronger than the 32 desire to express "preference" remains unclear. I do not think that the word “desire” is appropriate in this context.

Line 34: “easier repeatability” of what?

Line 48: Once more the vocabulary appear inappropriate: “authority and autonomy” in which context? In mate choice with respect to males? Even if this is the concept that the authors want to express, they should use other, more adequate, terms.

Line 49: “mate choice rights”, same as previous comment.

Lines 52-56: this part on the choice of “dominant genes” may be misleading. An individual during the choice of a preferred partner select a number of characteristics which are ultimately linked to the expression of a specific pattern of genes, but a preferred feature might be determined by the interaction among a set of genes and not necessarily all these genes will be dominant. I think that in this case the authors might be trying to say that these genes are “dominant” in the sense that their combined expression produce a particular phenotypic trait which is preferred by potential partner, but, once more I think it should be more carefully and clearly explained to avoid confusion with the paradigm “dominant” vs “recessive” when referring to genes.

Line 60: What does “group-disposed males” mean?

Line 61: “Ma” should be “Mate” I guess.

Line 63: Please explain what “positive courtship behaviors” means.

Lines 60-64: Instead of providing a simple list of examples of preferred characteristics, the authors should try and provide a bit of information the biological reasons for these choices, for example the fact that males across multiple taxa often prefer larger females because female body size often correlates with fecundity.

Line 65-66: Please clarify “a comprehensive selection of each other’s advantages”.

Lines 71-81: The authors should also discuss the critical point with no-choice tests regarding the fact that, in the experimentally confined environment of a laboratory, individuals may be forced to choose the presented potential partner. Thus, although I agree that the contemporary dichotomous choice may not be frequent in nature, the authors should stress that the no-choice tests should consider a sequential selection through subsequent random trials.

Lines 81-82: I wouldn’t speak of “expression of one's own wishes” when talking about mate selection in fishes.

Lines 84: “evident” instead of “obvious”.

Lines 85-86: Clarify “free-choice reproductive characteristics, with an obvious preference and willingness to choose a mate”. As already asked, what are “free-choice reproductive characteristics”? And why the preference should be “obvious”?

Line 89: As said above, the authors must clarify what are the “behavioral performance characteristics”.

A major point regarding the two experiments is that, in both cases, both males and females were able to interact, which means that the behaviour of the focal individual (males in experiment 1, females in experiment 2) will depend not only on the characteristics of the presented potential mate, but also on its behavioral interaction which might be a huge bias. For example, a male might be highly interested and could prefer a particular female, for example larger females, but if these females were not interested in mating and reject the approaching males, they could result as non-preferred females, simply because they were not willing to mate. Since the duration of the encounters was relative short (20 minutes), the interactions among males and females should be considered. A better experimental setup would have considered the frequency and intensity of male and female courtship/mating behaviour using fish lures instead of other individuals. Especially with such a low sample size.

Lines 100-104: further details about sampling are needed. Were the lab-reared males and females related? This is an important information that must be provided, since it is well known that relatedness can influence mate choice.

Lines 109: The authors could explain why the recording were acquired between 19:00 and 21:00, I am not an expert in the daily rhythm of this fish, but are they active in mating behaviour at night in the field? I guess that, if visual cues/signals are used in mate choice (body coloration, etc.), the best time to carry out the experiments should not be at night.

Lines 120, 135: I am a bit surprised to see such an extremely low sample size. Only 6 males and 6 females were tested? I think that this is the final sample size, it might be far too low to obtain solid and robust results.

Lines 120, 135: What do the authors mean with “healthy” males and females? How was the health condition measured?

Line 121: it is not clear how males build a nest. Aequidens should be cichlid species which lay their eggs on a surface, so I wonder how was structured the nest prepared by males. I also wonder the clear link between the building of the nest and how this indicates that the male has the ability and energy to choose a mate, are there any references that prove this evidence?

Lines 122-123: I would say “male acclimatisation”, “domestication” is a totally different process!

Line 126: The authors should describe in detail how to measure the “willingness to mate”. Providing references is not enough.

Line 126: “evident” instead of “obvious”.

Lines 129-130: It is not clear if a male was presented with different females, but this should not be the case since at lines 117-118 is reported that males and females were used only once. Furthermore, I wonder if all the behavioral analyses are based on only 2 preferred and 2 non-preferred females (which means a sample size of 4 in total, which would be even lower than the 6 individuals tested).

Line 136: once more, what are this alleged “nesting behaviours”?

Lines 137-139: I express the same perplexities as for experiment 1 (lines 129-130).

Line 147: “Behavioral type” remains unclear. The authors must anticipate which behaviours or courtship displays were considered/expected before the results.

Line 148: “a particular behavior” instead of “this behavior”.

Line 161: It is not clear which are these 20 fishes? Are they taken from those tested? But from the text it seems that a total of 24 individuals were tested (6 males and 6 females for each of the two experiments). The authors should clarify this point.

Line 181: Clarify “behavioral initiative”.

Figure 1 is impossible to read and, from what can be seen, it seems that tested fishes expressed a very low frequency and duration of behaviours, especially in panels C and D.

Lines 235-238: the preference for smaller females is quite awkward and unexpected.

Paragraph 3.2: This paragraph is redundant with respect to 3.1, the authors should try and condense both, since the experimental design is exactly the same.

Figure 5: same comment as for figure 1.

Paragraphs 3.1.7 and 3.2.3. are entangled. I would suggest the authors to move the tables of the PCA analyses in supplementary material.

Line 329: “self-presentation”?

Line 354: “higher” rather than “elevated”. However, since there was no statistical difference is not possible to say that testosterone levels were higher or more elevated in preferred males. They simply did not differ!

Line 369: Once more, I do not think that “the right to select” is an appropriate expression.

Line 369. “Parental care” instead of “spawn protection behaviors”.

Line 388: “mate selection” instead of “heterosexual selection”.

Lines 395-399: please change the sentence “the desire to express dislike was stronger than the desire to express like” with a more appropriate one when talking about mate choice in fishes.

Line 404: How was female attractiveness evaluated? Same for practical ability and physical strength of males. From the PCA paragraphs is not easy to understand these points.

Line 407: “benefits of the male to oneself”?

Lines 419-421: the proposed explanation is quite weak.

Lines 421-423: If the difference was not significant, than the testosterone levels between preferred and non-preferred were not different, not higher in preferred males.

Lines 423-425: unclear.

Lines 424-429: Why then the authors measured sex hormone levels if they did not add sex hormone levels to PCA analysis of preference characteristics? The following explanation (lines 427-429) is not satisfactory.

Lines 439-450: The premises on the preferences for larger body sizes, which makes sense, is not logically subsequent with the authors explanation that males A. rivulatus might have a low mate encounter rate, meaning that if a mate is rejected on a no-choice test, future mating opportunities may not be guaranteed and this implies that large body sizes may not be the standard for male A. rivulatus. Even if males have a low encounter rate, if they encounter a larger female, they should still prefer it! In fact, they should prefer even more a partner of higher reproductive value if these are difficult to encounter.

Lines 451-454: unclear

Lines 470: provide reference for Anders and Gunilla.

Line 472: “attach” in inappropriate.

Author Response

The response to reviewer #2 was uploaded as separate attachment.  

Reviewer 3 Report

The manuscript entitled as "Differences in Mate Choice Preference Characteristics of 2 Aequidens rivulatus" is an interesting manuscript showing different levels of paradigm of mate choice in fishes. However, there are certain concerns that I would like authors should addressed that in the manuscript:

  1. In the experimental design section authors should mention about the criteria of sexual maturity i.e., how they know that fishes are mature.
  2. In the line 61 and 62 authors took 20 females and males is it combined or 20 females and males separately. Please clear that.
  3. Whether age groups are same or different for female and males
  4. While using ANOVA authors should also used post-hoc test 
  5. Line 176 authors should provide reference of different courtship behavior in fishes.
  6. Size of the Figure all figures in the text should be large as legends on X and Y axis are not visible
  7. Line 402 and 403 authors should discuss about the pheromone level involved in courtship behavior
  8. Iine 45 and in reference add reference " Role of sexual selection in speciation in Drosophila"by Singh A and Singh BN, which discuss extensively about pre-copulatory sexual selection.

Author Response

The response to reviewer #3 was uploaded as separate attachment.  

Round 2

Reviewer 1 Report

The paper is a revision of a previously reviewed paper. It is much improved and the authors have acted upon the comments and suggestions appropriately. The only problem I found was at the end of the literature cited where there are two references listed and it is unclear whether the authors wanted to include them in the text or to delete them.

Author Response

Point 1: The paper is a revision of a previously reviewed paper. It is much improved and the authors have acted upon the comments and suggestions appropriately. The only problem I found was at the end of the literature cited where there are two references listed and it is unclear whether the authors wanted to include them in the text or to delete them.

Response 1: Thanks for your affirmation of our revised manuscript. We are sorry for the confusion. The two references were included and cited in the revised manuscript as Line 549-552 and Line 552-554: “This can be explained by the fact that female behavioral intensities were less intense than those for males in male mate choice. Female behaviors may not only be a signal of comfort, a way to assess a mate, but may also have the effect of inducing male courtship [75].” And “Therefore, males pay more attention to female behaviors when choosing a mate. On the other hand, in female mate choice, the intensities of male behaviors represent some form of female harassment [76].”

  1. Barbosa, M.; Magurran, A.E. Female mating decisions: maximizing fitness? J Fish Biol 2006, 68, 163-1661.
  2. Medina, L.M.; Garcia, C.M.; Urbina, A.F.; Manjarrez, J.; Moyaho, A. Female vibration discourages male courtship be-haviour in the Amarillo fish (Girardinichthys multiradiatus). Behav Process 2013, 100, 163-168.

Reviewer 2 Report

The authors have put a lot of effort in responding to most of the comments on the previous version and I appreciate it. However, I still believe that the fact that the authors did not consider interactions between males and females represents a considerable bias and the authors response is not satisfactory.

Another major point that remains is the fact that "All fish were sorted from the same group and shared the same genetic and environmental background". Does this mean that the tested fish were siblings? Or some of them were while others were not? This is potentially an even greater bias and the authors should carefully reflect about their experimental design. 

As regards the explanation of the authors to justify the recording acquired between 19:00 and 21:00, they cite https://www.seriouslyfish.com/species/andinoacara-rivulatus/ which does not seem to be a reliable and scientific support for their choice.

The language might still need an extensive editing since in many parts sound awkward or inadequate;

e.g. "predominance and activity in mate choice" is a little improvement compared to "authority and autonomy", "mate choice rights" or "right to select".

"intensive courtship behaviors" instead of "intensively courtship behaviors".

In my opinion "the expression of one's own willingness" is still inadequate when referring to fish as well as "self-display" instead of "self-presentation".

"benefits of the male to female" should be "benefits provided by males".
